# Hysteresis Modeling and Compensation for a Fast Piezo-Driven Scanner in the UAV Image Stabilization System

**Jinlei Lu** , **Jun Wang** * , **Yuming Bo and Xianchun Zhang**

School of Automation, Nanjing University of Science and Technology, Nanjing 210094, China; jinlei_lu@njust.edu.cn (J.L.); byming@njust.edu.cn (Y.B.); zxc4313075@njust.edu.cn (X.Z.)
* Correspondence: wangjun1125@njust.edu.cn

**Abstract:** The fast piezo-driven scanner (FPDS) compensates for vibrations in the unmanned aerial vehicle (UAV) image stabilization system. However, the hysteresis nonlinearity reduces the positioning accuracy of the FPDS. To address this challenge, this paper presents a novel weighted polynomial modified Bouc–Wen (WPMBW) model cascaded with a linear dynamic model to describe counterclockwise, asymmetric, and rate-dependent hysteresis loops of an FPDS. The proposed approach utilizes the weighted polynomial function to describe the asymmetric characteristic and the linear dynamic model to capture the rate-dependent behavior. By modifying the last two terms in the classical Bouc–Wen (CBW) model, the modified BW model directly characterizes the counterclockwise hysteresis loops with fewer parameters, circumventing the algebraic-loop problem that arises in the inverse CBW model. The pseudorandom binary sequence (PRBS) input is employed to decouple the linear dynamic model from the WPMBW model. The sinusoidal input is then applied to stimulate the hysteresis phenomenon, and the parameters of the WPMBW model are estimated by the particle swarm optimization (PSO) toolbox. Experimental results on a commercial FPDS show that the proposed model is superior to the CBW and traditional asymmetric BW models in modeling accuracy and feedforward hysteresis compensation.

**Keywords:** fast piezo-driven scanner; UAV image stabilization system; hysteresis nonlinearity; Bouc–Wen model; feedforward hysteresis compensation



## 1. Introduction

Unmanned aerial vehicles (UAVs) have achieved rapid advances in recent years, opening up a broad range of applications, such as precision agriculture [1], wireless communication [2], mining [3], and disaster management [4]. Typically, UAV-based applications require a robust image stabilization system to capture high-quality photos for detailed analysis and accurate decision-making [5]. However, during UAV flight, the camera experiences vibrations that adversely affect the quality of captured images. The fast piezo-driven scanner (FPDS), known for fast response speed, compact size, low power consumption, and high positioning precision, can compensate for vibrations in the UAV imaging system [6]. Unfortunately, an FPDS is hindered by hysteresis, a nonlinear phenomenon caused by the inverse piezoelectric effect that considerably diminishes positioning accuracy, resulting in undesirable image distortions [7]. Consequently, hysteresis modeling and compensation are crucial for realizing the full potential of an FPDS in the UAV image stabilization system.

Several compensation algorithms have been explored to mitigate the hysteresis phenomenon of an FPDS, which can be categorized into two primary groups: hysteresis model-free and hysteresis model-based approaches [8]. Concerning the hysteresis model-free strategy, the hysteresis is regarded as a bounded disturbance to the nominal system [9]. Different advanced feedback controllers, such as sliding mode control [10,11], adaptive control [12,13], and fuzzy control [14,15], have been developed to alleviate the hysteresis effect. On the other hand, the hysteresis model-based approach aims to establish the hysteresis model as accurately as possible [16,17]. The inverse model based on the obtained

hysteresis model is utilized as a feedforward controller to eliminate the hysteresis effect. The performance of the inverse hysteresis feedforward controller relies heavily on the precision of the identified hysteresis model. Hence, an accurate hysteresis model is of paramount importance.

For this purpose, various types of phenomenological and mathematical hysteresis models have been proposed. The phenomenological hysteresis modeling is mainly based on a thermodynamic framework. Zhou et al. [18] revealed frequency-dependent polarization and strain characteristics of soft lead zirconate titanate (PZT) piezoelectric ceramics under electric field loading. Additionally, they observed an elevated coercive field with increasing loading frequency, which was tentatively ascribed to rate-induced influences in the domain-switching mechanism. Delibas et al. [19] proposed a three-dimensional micromechanical model for simulating the rate-dependent behavior of specific perovskite tetragonal piezoelectric material, and incorporating a probability function allowed for a more accurate description of hysteresis curves. Other works can be found in [20–22].

The mathematical hysteresis models can be classified into three main categories [23]. The first category contains operator-based (OPRB) hysteresis models, such as the Preisach model [24], Prandtl–Ishlinskii (PI) model [25], and Krasnosel'skii–Pokrovskii (KP) model [26]. The second category includes differential equation-based (DEB) hysteresis models, for instance, the Duhem model [27], backlash-like model [28], and Bouc–Wen (BW) model [29]. The third category comprises artificial intelligent-based (AIB) hysteresis models using a support vector machine [30] or neural network [31]. Compared to the OPRB and AIB hysteresis models, the DEB hysteresis models demand less computational power because of the simpler mathematical structures and fewer parameters. In particular, the classical BW (CBW) model has attracted much attention since it employs only a first-order nonlinear differential equation. Furthermore, the CBW model is capable of incorporating real-life physical properties into mathematical problems [32].

The CBW model is limited to simulating symmetric and rate-independent hysteresis loops, whereas in practice, the hysteresis loops of an FPDS often exhibit counterclockwise, asymmetric, and rate-dependent characteristics [33]. Therefore, several attempts have been made to modify the CBW model to capture the hysteresis loops of an FPDS more precisely. For instance, to describe asymmetric hysteresis loops, Zhu et al. [34] incorporated an asymmetric formula into the CBW model, while Wang et al. [35] augmented the CBW model with a polynomial-based non-lagging component. To capture rate-dependent hysteresis loops, Zhu et al. [36] integrated a frequency factor into the CBW model, while Kang et al. [37] introduced two fractional operators into the CBW model. Additionally, some recent works adopted the Hammerstein structure cascading the rate-independent hysteresis component, such as the CBW model, with the rate-dependent component, such as the transfer function, to extend the hysteresis model over a wide range of frequencies [38–40].

Although impressive outcomes have been achieved in previous works, they are still not entirely satisfactory, particularly in the investigation of the directions of hysteresis loops. To describe counterclockwise hysteresis loops, the CBW model is generally expressed as $y(t) = du(t) - h[u(t)]$, where $u(t)$ and $y(t)$ refer to the input voltage and output displacement, respectively, $d$ denotes the piezoelectric coefficient, and $h[u(t)]$ represents the solution of the CBW model. The inverse multiplicative structure (IMS) [29] is commonly employed to construct the inverse CBW feedforward compensator, i.e., $u(t) = 1/d(y(t) + h[u(t)])$. However, to calculate $u(t)$, we need the value of $h[u(t)]$, but to determine $h[u(t)]$, we need the value of $u(t)$. This circular dependency makes it challenging to find a unique solution for $u(t)$ and $h[u(t)]$ simultaneously, leading to the algebraic-loop problem. To address this issue, a constant time delay or a low pass filter is typically incorporated into the IMS [41]. Nevertheless, this operation introduces additional compensation errors.

Based on the Hammerstein structure, this paper proposes a novel weighted polynomial modified BW (WPMBW) model cascaded with a linear dynamic model to describe counterclockwise, asymmetric, and rate-dependent hysteresis loops observed in an FPDS. The introduced weighted polynomial function describes the asymmetric characteristics,

and the linear dynamic model captures the rate-dependent behavior. The MBW model directly describes counterclockwise hysteresis loops with fewer parameters by modifying the last two terms of the CBW model, thereby avoiding the algebraic-loop issue that arises in the inverse CBW model. The relationship between the parameters of the WPMBW model and the shape of hysteresis loops is studied. In particular, the output bound of the MBW model is derived. Taking inspiration from the decoupling identification research of the Hammerstein system [42], the linear dynamic model is stimulated with the pseudorandom binary sequence (PRBS) input, and the linear parameters are estimated using the MATLAB system identification toolbox. Meanwhile, a sinusoidal signal is employed to excite the hysteresis phenomenon, and the particle swarm optimization (PSO) toolbox [43] is used to determine the parameters of the WPMBW model. Using the identified WPMBW model, the inverse WPMBW feedforward controller is also implemented to mitigate the hysteresis effect. Comparative experiments conducted on a commercial FPDS demonstrate that the proposed model provides better modeling accuracy than the CBW model [29] and traditional asymmetric BW model [36], achieving superior hysteresis compensation results.

The remainder of this paper is organized as follows. Section 2 presents the hysteresis modeling of an FPDS. The characteristics of the proposed WPMBW model cascaded with a linear dynamic model are discussed in Section 3. The hysteresis identification, verification, and compensation experiments are carried out in Section 4 to validate the effectiveness of the proposed model. Finally, Section 5 concludes this paper.

## 2. Hysteresis Modeling of an FPDS

This section introduces the hysteresis properties of an FPDS, followed by the development of the proposed WPMBW model cascaded with a linear dynamic model.

### 2.1. Hysteresis Characterization

As depicted in Figure 1, the FPDS acts as an active isolation device to compensate for disturbances during the UAV flight [44]. Prior to modeling the hysteresis nonlinearity of the FPDS, it is imperative to analyze the impact of input amplitudes and frequencies on the shape of hysteresis loops. Therefore, sinusoidal inputs with varying amplitudes and frequencies are applied to the *x*-axis of the FPDS, as shown in Figure 2. As can be seen, the hysteresis loops of the FPDS include two phases, namely the ascending and descending phases. Moreover, the direction of hysteresis loops from the ascending phase to the descending phase is counterclockwise. Figure 2a illustrates the amplitude-dependent property of hysteresis loops because the width between the ascending and descending phases increases with rising input amplitudes. Figure 2b reveals the frequency-dependent characteristic of hysteresis loops as the hysteresis effects between the ascending and descending phases become more pronounced with increasing input frequencies.

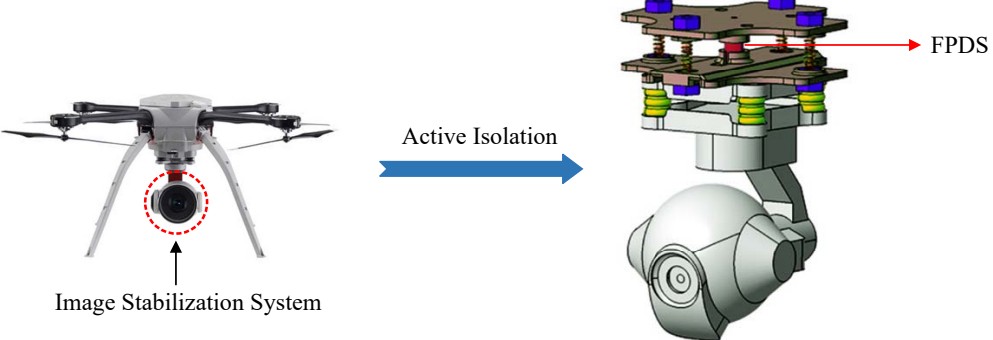

**Figure 1.** Schematic diagram of an FPDS in the UAV image stabilization system.

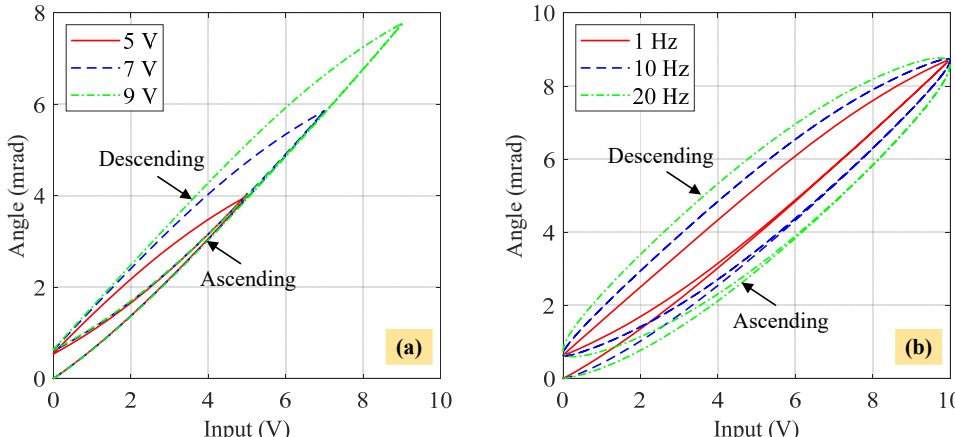

**Figure 2.** Counterclockwise hysteresis loops of the *x*-axis of an FPDS under varying sinusoidal inputs $u(t) = 0.5A[1 - \cos(2\pi ft)]$ (V). (**a**) Amplitude-dependent hysteresis loops under increasing amplitudes where $A = 5, 7, 9$ V, and $f = 1$ Hz. (**b**) Frequency-dependent hysteresis loops under increasing frequencies where $f = 1, 10, 20$ Hz, and $A = 10$ V.

In general, hysteresis loops can be decomposed by subtracting a linear component $l(t)$ from a hysteretic component $h(t)$ [36]. Figure 3 shows an example of the hysteresis loop decomposition result where *a* and *b* are inflection points during the ascending and descending phases. Figure 3b indicates the asymmetric property of hysteresis loops since the absolute value of *a* and *b* is unequal. To sum up, the hysteresis loops of an FPDS exhibit counterclockwise, asymmetric, amplitude-dependent, and rate-dependent features.

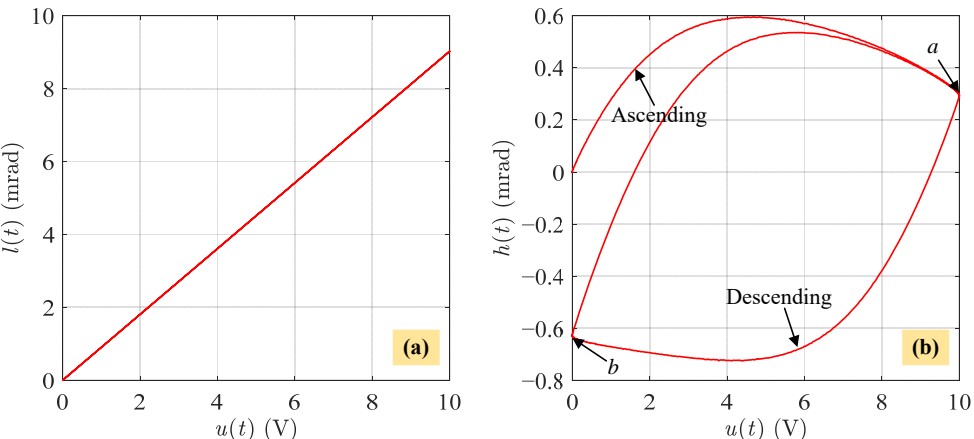

**Figure 3.** An example of the hysteresis loop decomposition result under $u(t) = 5[1 - \cos(2\pi t)]$ (V). (**a**) The linear component. (**b**) The hysteretic component.

### 2.2. The CBW Model

The CBW model, for its simplicity of expression and capability of describing extensive hysteretic systems, has been widely employed to represent the hysteresis phenomenon of piezoelectric actuators. In this work, the hysteresis dynamics of an FPDS with the CBW model can be expressed as follows:

$$y(t) = du(t) - h(t) \tag{1}$$

$$\dot{h}(t) = \alpha \dot{u}(t) - \beta |\dot{u}(t)| h(t) - \gamma \dot{u}(t) |h(t)| \tag{2}$$

where $u(t)$ represents the input voltage to the FPDS; $y(t)$ denotes output angles of the FPDS; $d > 0$ is the piezoelectric coefficient; and $\alpha$, $\beta$, and $\gamma$ are parameters used to determine the magnitude and shape of the CBW hysteresis operator *h*. The mathematical and physical

consistency of the CBW model was comprehensively discussed in [45,46]. It has been proven that if $\alpha$, $\beta$, and $\gamma$ satisfy,

$$\alpha > 0, \beta + \gamma > 0, \beta - \gamma \geq 0 \tag{3}$$

then the CBW model exhibits properties of the bounded-input-bounded-output (BIBO) stability, passivity, and inherent thermodynamic admissibility.

**Lemma 1.** *If $\alpha$, $\beta$, and $\gamma$ satisfy (3), then (2) can only produce stable clockwise hysteresis loops [46].*

Therefore, the subtraction operation on the right-hand side of (1) is introduced to generate the counterclockwise hysteresis loops when modeling an FPDS with the CBW model. Since the inverse model of (2) cannot be solved explicitly, the IMS is adopted to construct the inverse CBW model, i.e., $u(t) = 1/d(y(t) + h[u(t)])$. However, the algebraic-loop problem arises in the inverse CBW model. This issue motivates us to modify the CBW model to avoid the algebraic-loop problem when solving its inverse model.

*2.3. Proposed MBW Model*

To address the algebraic-loop issue, we first transform (2) into the following form:

$$\dot{w}(t) = \rho(\dot{u}(t) + \lambda u(t)|\dot{w}(t)| - \mu w(t)|\dot{w}(t)|) \tag{4}$$

where parameters $\rho$, $\lambda$, and $\mu$ are positive constants used to decide the magnitude and shape of the MBW hysteresis operator $w$. Compared to (2), the second and third terms on the right-hand side of (2) are modified. It is notable that the right-hand side of (4) also contains $\dot{w}(t)$. Substituting $|\dot{w}(t)| = \dot{w}(t)\mathrm{sgn}(\dot{w}(t))$ into (4), it yields

$$\dot{w}(t) = \dot{u}(t)\frac{\rho}{1 - \rho\lambda u(t)\mathrm{sgn}(\dot{w}(t)) + \rho\mu w(t)\mathrm{sgn}(\dot{w}(t))} \tag{5}$$

If the hysteresis behavior of a system displays saturation characteristics, wherein the output remains constant as the input increases, then (5) can be cascaded with a saturation operator, such as the arctangent-polynomial operator [47]. However, since the hysteresis dynamics of an FPDS do not exhibit saturation features, we only need to consider the scenario that the sign of $\dot{u}(t)$ always matches that of $\dot{w}(t)$. In other words, $\mathrm{sgn}(\dot{w}(t))$ can be replaced by $\mathrm{sgn}(\dot{u}(t))$ in (5). To this end, the proposed MBW model is derived as follows:

$$\dot{w}(t) = \begin{cases} \dot{u}(t)\frac{\rho}{1 - \rho\lambda u(t) + \rho\mu w(t)}, & \text{if } \dot{u}(t) \geq 0 \\ \dot{u}(t)\frac{\rho}{1 + \rho\lambda u(t) - \rho\mu w(t)}, & \text{if } \dot{u}(t) \leq 0 \end{cases} \tag{6}$$

The denominator in (6) can be regarded as a feedback mechanism that causes $w(t)$ to regulate its own growth rate, thereby maintaining the stability of the system. According to (6), the computation of $w(t)$ relies not only on the current input voltage but also on its previous values. Therefore, (6) effectively explains the nonlocal memory of hysteresis loops.

Compared to the CBW model (1) and (2), our proposed MBW model (6) can directly describe counterclockwise hysteresis loops with fewer parameters. Moreover, (6) has a direct inverse model, which solves the algebraic-loop problem. The counterclockwise property and inverse model of (6) will be studied in the next section.

*2.4. Proposed WPMBW Model Cascaded with a Linear Dynamic Model*

Inspired by the work in [37], to capture the asymmetric hysteresis loops of the FPDS, the weighted polynomial function is cascaded with the MBW model (6) as follows:

$$v(t) = \sum_{i=1}^{L} c_i(w(t))^i \tag{7}$$

where $c_i$ and $L$ are the coefficient and order of the polynomial function $v(t)$, respectively. Combining (6) with (7), the asymmetric WPMBW hysteresis model is obtained.

To describe the rate-dependent hysteresis phenomenon, a linear dynamic model is cascaded with (6) and (7) as follows:

$$\sum_{i=0}^{n} a_{n-i} y^{(n-i)}(t) - \sum_{j=0}^{m} b_{m-j} v^{(m-j)}(t) = 0 \tag{8}$$

where $a_{n-i}$ and $b_{m-j}$ represent the coefficient of the $(n-i)$-th derivative of $y(t)$ and $(m-j)$-th derivative of $u(t)$, respectively.

To this end, the proposed WPMBW model cascaded with a linear dynamic model is synthesized by (6)–(8), as shown in Figure 4. The WPMBW model captures the counterclockwise, asymmetric, and amplitude-dependent hysteresis loops, whereas the linear dynamic model describes the rate-dependent features of hysteresis loops. Figure 4 also depicts that the proposed model (6)–(8) is formulated based on the Hammerstein structure, wherein the nonlinear component is cascaded with the linear component. Thus, when identifying the proposed model, the PRBS signal can be applied to decouple the WPMBW model from the linear dynamic model, thereby considerably simplifying the identification complexity. The characteristics of the proposed model will be comprehensively discussed in the following section.

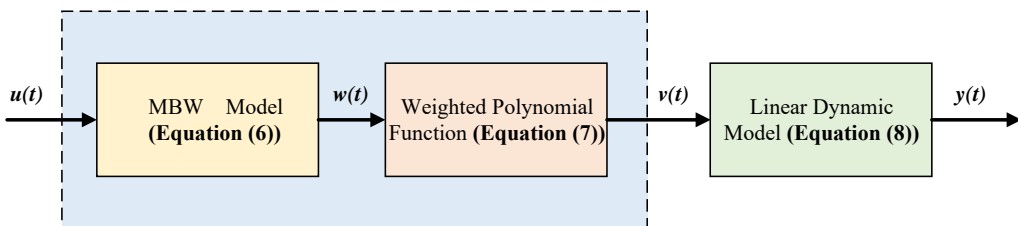

**Figure 4.** Schematic diagram of the proposed WPMBW model cascaded with a linear dynamic model.

## 3. Characteristics of the WPMBW Model

This section highlights the relationship between the counterclockwise, asymmetric, amplitude-dependent, and rate-dependent features of the proposed WPMBW model cascaded with a linear dynamic model and its model parameters. Additionally, the inverse WPMBW model is established.

### 3.1. Counterclockwise Characteristics

To investigate the direction of hysteresis loops produced by (6), the second derivative is derived as follows:

$$\frac{\mathrm{d}^2 w(t)}{\mathrm{d}u(t)^2} = \begin{cases} \dot{u}(t)\frac{\rho^2(-\rho\mu+\lambda-\rho\lambda^2 u(t)+\rho\lambda\mu w(t))}{(1-\rho\lambda u(t)+\rho\mu w(t))^3}, & \text{if } \dot{u}(t) \geq 0 \\ \dot{u}(t)\frac{\rho^2(\rho\mu-\lambda-\rho\lambda^2 u(t)+\rho\lambda\mu w(t))}{(1+\rho\lambda u(t)-\rho\mu w(t))^3}, & \text{if } \dot{u}(t) \leq 0 \end{cases} \tag{9}$$

Since the signs of $\dot{u}(t)$ and $\ddot{w}(t)$ remain the same, one can obtain

$$\frac{\rho^2}{(1-\rho\lambda u(t)+\rho\mu w(t))^3} > 0 \tag{10}$$

and

$$\frac{\rho^2}{(1+\rho\lambda u(t)-\rho\mu w(t))^3} > 0 \tag{11}$$

Combining (10) and (11), the sign of (9) relies on the signs of the following two terms:

$$\phi(t) = -\rho\mu + \lambda - \rho\lambda^2 u(t) + \rho\lambda\mu w(t) \tag{12}$$

$$\chi(t) = \rho\mu - \lambda - \rho\lambda^2 u(t) + \rho\lambda\mu w(t) \tag{13}$$

Consequently, the proposed MBW model (6) can produce counterclockwise hysteresis loops when $\phi(t) \geq 0$ and $\chi(t) \leq 0$. On the contrary, clockwise hysteresis loops can be generated when $\phi(t) \leq 0$ and $\chi(t) \geq 0$. That is to say, the proposed MBW model can describe either the counterclockwise or clockwise hysteresis loops by varying parameters, as demonstrated in Figure 5.

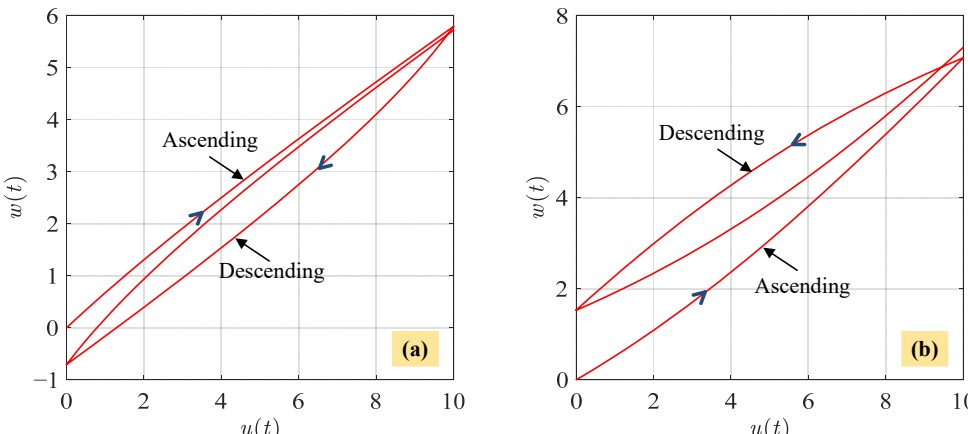

**Figure 5.** Different directions of hysteresis loops generated by the MBW model (6) under $u(t) = 5[1 - \cos(2\pi t)]$. (**a**) The clockwise hysteresis loop with $\rho = 0.7, \lambda = 0.3$, and $\mu = 0.6$. (**b**) The counterclockwise hysteresis loop with $\rho = 0.5, \lambda = 0.4$, and $\mu = 0.45$.

In this study, we only focus on the counterclockwise hysteresis loops generated by (6) because our primary goal is to model the FPDS. From $\phi(t) \geq 0$ and $\chi(t) \leq 0$, one can obtain $\phi(t) - \chi(t) \geq 0$. Substituting (12) and (13) into this inequality, we have

$$\lambda \geq \rho\mu \tag{14}$$

It should be noted that (14) is a critical condition for counterclockwise hysteresis loops generated by (6). On the other hand, we can infer that $\lambda \leq \rho\mu$ for clockwise hysteresis loops. Moreover, it is possible to explore the output constraints of the MBW model (6) under $T$-period inputs.

**Definition 1** ([48,49])**.** *Suppose that $u(t)$ is continuous in $[0, +\infty)$ and periodic with a period of $T$ ($T > 0$). There exists a peak time $T^+$ and an integer $m$ within the period $[0, T)_m = [mT, mT + T)$ such that $\dot{u}(t) > 0$ for $t \in (0, T^+)_m$ and $\dot{u}(t) < 0$ for $t \in (T^+, T)_m$. $U_{\min} = u(mT)$ and $U_{\max} = u(mT + T^+)$ denote the minimum and maximum values of the input $u(t)$, respectively. This kind of ascending-descending input signal shape is regarded as a $T$-period input, as illustrated in Figure 6.*



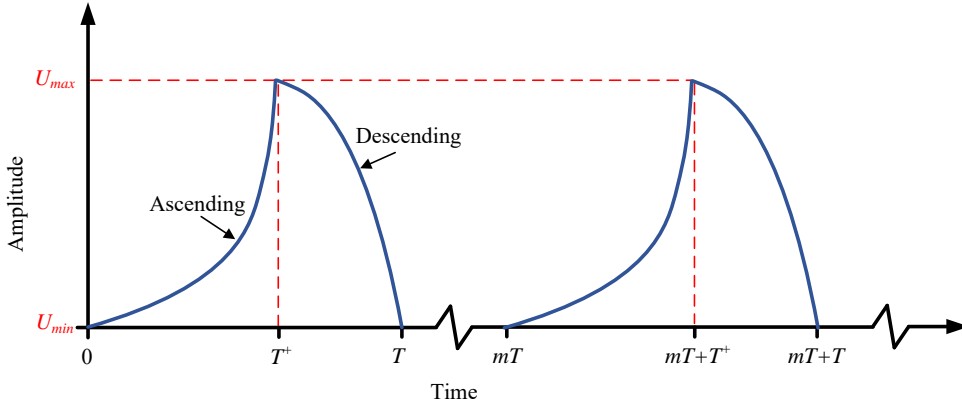

**Figure 6.** Schematic diagram of the *T*-period input signal.

If $\phi(t) \geq 0$ and $\chi(t) \leq 0$, we have

$$\begin{cases} w(t) \geq \frac{\lambda}{\mu}u(t) - \frac{\lambda - \rho\mu}{\rho\mu\lambda}, & \text{if } \dot{u}(t) > 0 \\ w(t) \leq \frac{\lambda}{\mu}u(t) + \frac{\lambda - \rho\mu}{\rho\mu\lambda}, & \text{if } \dot{u}(t) < 0 \end{cases} \tag{15}$$

Since $u(t)$ is a continuous *T*-period input, the output bounds of the MBW model (6) are obtained as follows:

$$\frac{\lambda}{\mu}u(t) - \frac{\lambda - \rho\mu}{\rho\mu\lambda} \leq w(t) \leq \frac{\lambda}{\mu}u(t) + \frac{\lambda - \rho\mu}{\rho\mu\lambda} \tag{16}$$

Furthermore, (16) reveals how the parameters influence the shape of MBW hysteresis loops. To facilitate discussion, we denote the lower bound of $w(t)$ as $w_l(t)$ and the upper bound as $w_u(t)$. Notably, $w_l(t)$ and $w_u(t)$ are both linear functions of $u(t)$, i.e., $w_l(t) = k_w u(t) - b_w$ and $w_u(t) = k_w u(t) + b_w$, where the slope $k_w = \lambda/\mu$, and the intercept $b_w = (\lambda - \rho\mu)/\rho\mu\lambda$. Figure 7 depicts different shapes of MBW hysteresis loops by varying $k_w$ and $b_w$. As shown in Figure 7a, the tilt angle of MBW hysteresis loops can be determined by the slope $k_w$. The greater the value of $k_w$, the more pronounced the tilt angle of hysteresis loops. From Figure 7b, it can be observed that the intercept $b_w$ controls the width of MBW hysteresis loops. The higher the value of $b_w$, the larger the width of hysteresis loops. Therefore, based on the above analysis, the relationship between the parameters and the shape of MBW hysteresis loops has been established. It also indicates that the proposed MBW model is flexible to model extensive shapes of hysteresis loops by adjusting the parameters.

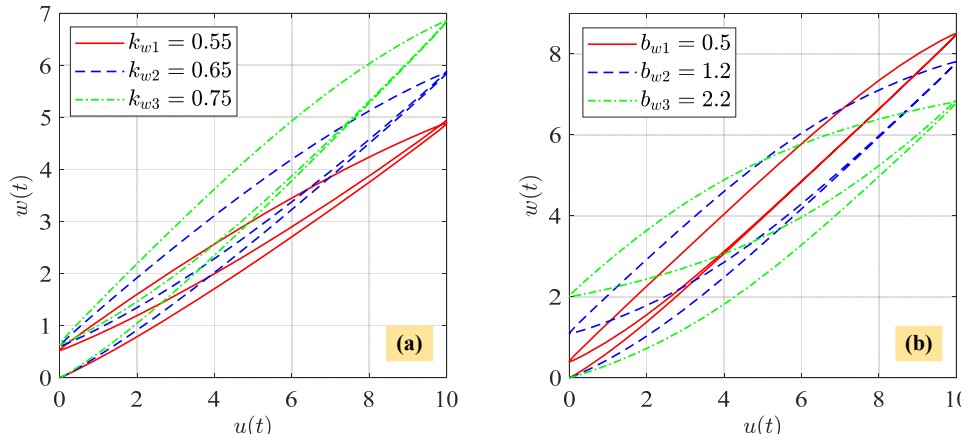

**Figure 7.** Different shapes of MBW hysteresis loops by varying $k_w$ and $b_w$ under $u(t) = 5[1 - \cos(2\pi t)]$. (**a**) $k_w$ increases while $b_w$ remains unchanged. (**b**) $b_w$ increases while $k_w$ remains unchanged.

### 3.2. Asymmetric Characteristics

The weighted polynomial function captures the asymmetric characteristic. Figure 8 intuitively presents the impact of different orders of $v(t)$ on the asymmetric feature. When $w(t)$ is positive, the weighted polynomial function $v(t)$ of second-order and above can produce asymmetric behavior. The asymmetric features become more prominent with increasing orders of the weighted polynomial function. Figure 9 demonstrates that the proposed WPMBW model can produce asymmetric hysteresis loops.

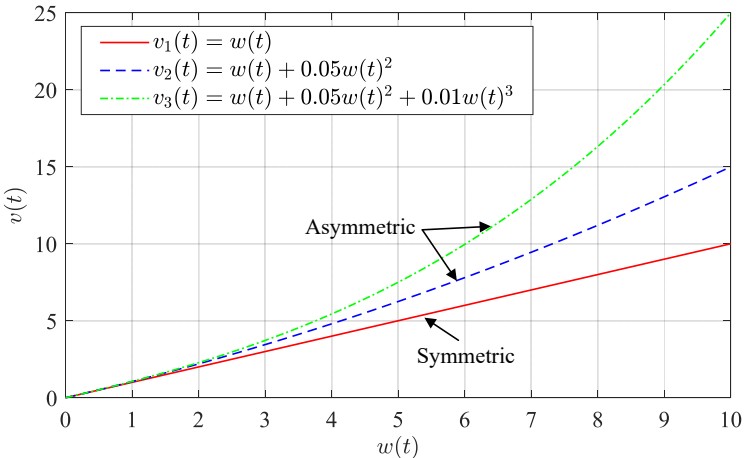

**Figure 8.** The relationship between different orders of $v(t)$ and the asymmetric characteristic.

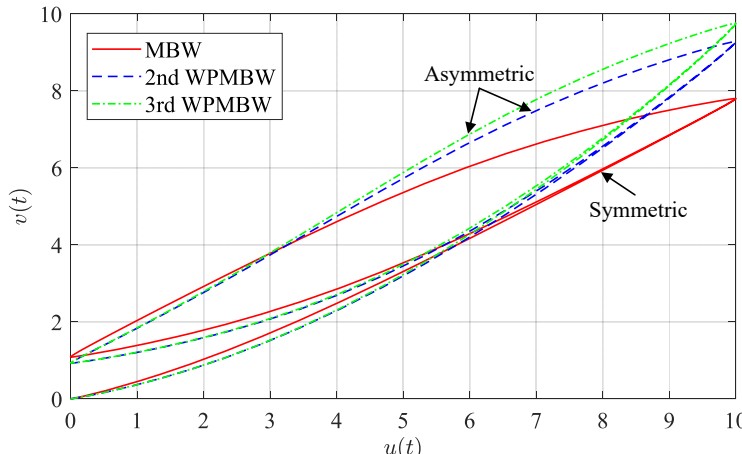

**Figure 9.** Comparison of hysteresis loops generated by the MBW and WPMBW models under $u(t) = 5[1 - \cos(2\pi t)]$, where the parameters of the MBW model are $\rho = 0.45$, $\lambda = 0.8$, and $\mu = 0.9$; the second-order weighted polynomial function is expressed as $v(t) = 0.8w(t) + 0.05w(t)^2$; and the third-order weighted polynomial function is formulated as $v(t) = 0.8w(t) + 0.05w(t)^2 + 0.001w(t)^3$.

### 3.3. Amplitude-Dependent and Rate-Dependent Characteristics

Figure 10a demonstrates that the hysteresis loops produced by the WPMBW model are amplitude-dependent. The WPMBW hysteresis loops become wider with increasing input amplitudes, which aligns with the hysteresis phenomenon observed in FPDS experiments (see Figure 2a). In Figure 10b, it is shown that the WPMBW hysteresis loops remain unchanged with increasing input frequencies, revealing that the WPMBW hysteresis loops are rate-independent.

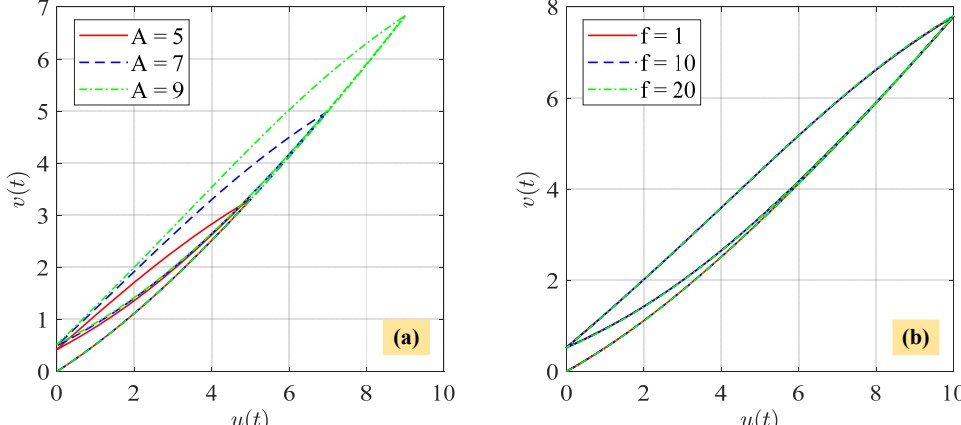

**Figure 10.** Different hysteresis loops generated by the WPMBW model under $u(t) = 0.5A[1 - \cos(2\pi f t)]$, where $\rho = 0.6, \lambda = 0.7, \mu = 0.8, c_1 = 0.8$, and $c_2 = 0.02$. (**a**) The input amplitude increases while the frequency remains unchanged. (**b**) The input frequency increases while the amplitude remains unchanged.

However, the hysteresis loops of an FPDS are dependent on the input frequencies (see Figure 2b). To address this issue, the linear dynamic model (8) is cascaded with the WPMBW model. As shown in Figure 11, the proposed WPMBW model cascaded with a linear dynamic model can describe rate-dependent hysteresis loops. The hysteresis effect becomes more pronounced with higher input frequencies, which is the same conclusion drawn from FPDS experiments.

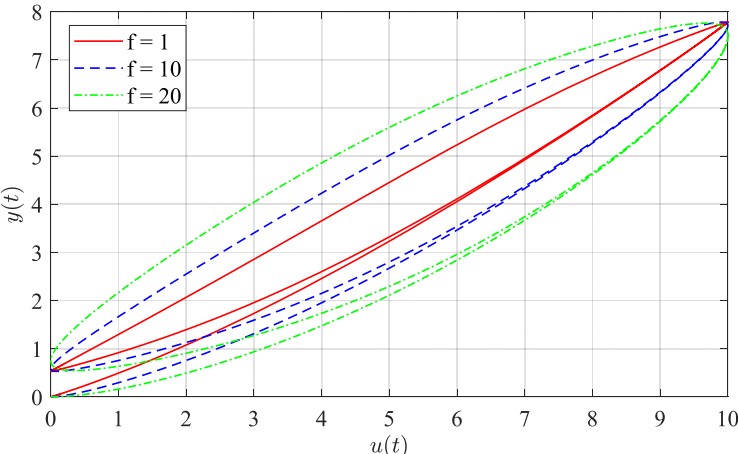

**Figure 11.** Different hysteresis loops generated by the WPMBW model cascaded with the second-order linear dynamic model under $u(t) = 5[1 - \cos(2\pi f t)]$, where $\rho = 0.6, \lambda = 0.7, \mu = 0.8, c_1 = 0.8$, $c_2 = 0.02, \xi = 0.8$, and $\omega_n = 200\pi$.

### 3.4. Inverse WPMBW Model

In addition to the counterclockwise, asymmetric, amplitude-dependent, and frequency-dependent features, the proposed WPMBW model cascaded with a linear dynamic model offers promising potential in hysteresis compensation thanks to its Hammerstein structure.

The order of the weighted polynomial function should be chosen based on a trade-off between the model accuracy and the identification complexity. In this work, a second-order weighted polynomial function, i.e., $v(t) = c_1 w(t) + c_2 w(t)^2$, is employed to describe the asymmetric characteristic. Thus, the inverse weighted polynomial function is calculated as follows:

$$w(t) = \frac{-c_1 + \sqrt{c_1^2 + 4c_2 v(t)}}{2c_2} \tag{17}$$

The negative root is ignored since $w(t) \geq 0$. It is worth noting that the roots of the third-order weighted polynomial function can be calculated using the Cardano formula. For higher-order weighted polynomial functions, the Newton iteration method can be utilized to find the roots. Previous works have demonstrated that the second-/third-order weighted polynomial function is sufficient to specify the piezoelectric asymmetric behavior [33,35,37].

Since the signs of $\dot{u}(t)$ and $\dot{w}(t)$ remain the same, based on (6), the inverse MBW model can be directly obtained as follows:

$$\dot{u}(t) = \begin{cases} \dot{w}(t)\frac{1-\rho\lambda u(t)+\rho\mu w(t)}{\rho}, & \text{if } \dot{w}(t) \geq 0 \\ \dot{w}(t)\frac{1+\rho\lambda u(t)-\rho\mu w(t)}{\rho}, & \text{if } \dot{w}(t) \leq 0 \end{cases} \tag{18}$$

Thus, by cascading (17) with (18), the inverse WPMBW model is obtained, as illustrated in Figure 12. Notably, the proposed WPMBW model provides a direct inverse model, avoiding the algebraic-loop problem that arises from the inverse CBW model. After the parameters of the WPMBW model are identified, the inverse WPMBW model can be constructed using (17) and (18), which makes it possible to implement the hysteresis compensator in a simple manner.

**Figure 12.** Schematic diagram of the inverse WPMBW model.

## 4. Hysteresis Identification, Verification, and Compensation

In this section, the parameters of the proposed WPMBW model cascaded with a linear dynamic model are identified on a commercial FPDS system. Furthermore, the effectiveness of the proposed model is verified through comparative experiments.

### 4.1. Experimental Setup

In this work, the two-degree-of-freedom (2-DOF) FPDS (model: P-T04K010), supplied by Physik Instrumente (PI) Shanghai branch, is employed to validate the proposed WPMBW model cascaded with a linear dynamic model. The FPDS was customized and developed based on the S-335 platform (https://www.physikinstrumente.com/en/products/nanopositioning-piezo-flexure-stages/piezo-flexure-tilting-mirrors/s-335-piezo-tiptilt-platform-300711 (accessed on 1 September 2019)) by PI Shanghai branch. The FPDS shares certain core components with the S-335 platform. The FPDS comprises four identical piezoelectric stack actuators (PSAs) and embedded SGS resistive sensors. Each pair of PSAs forms a push-pull motion to realize the fast scan function. The closed-loop maximum scan angle of the FPDS is ten mrad. In the UAV image stabilization system, the FPDS can act as an active isolation device to compensate for disturbances during the flight.

Figure 13 shows the experimental hardware connection. The target computer is composed of a PC104 controller (model: LX3160) based on the VxWorks operating system and a data sampling board (model: ADT800) with 16-bit A/D and 12-bit D/A converters, both from Shengbo Co. The programs are developed using the Tornado software on the host computer and then downloaded to the target computer. The control box (model: E517), manufactured by PI Co., contains signal conditional and voltage amplifier modules. The output angles of the FPDS are measured by its embedded capacitive sensors, which are then filtered through the signal conditioner. The filtered signals are converted to digital form by the A/D channel of the data sampling board and sent to the controller. Meanwhile,

the input command is generated by the controller and then converted to analog form by the D/A channel of the data sampling board. Subsequently, the command is magnified ten times by the voltage amplifier to drive the FPDS. The input voltage range is 0–10 V. Since the mechanical design completely decouples the two axes (*x*-axis and *y*-axis) of the FPDS, only the *x*-axis is examined in this study.

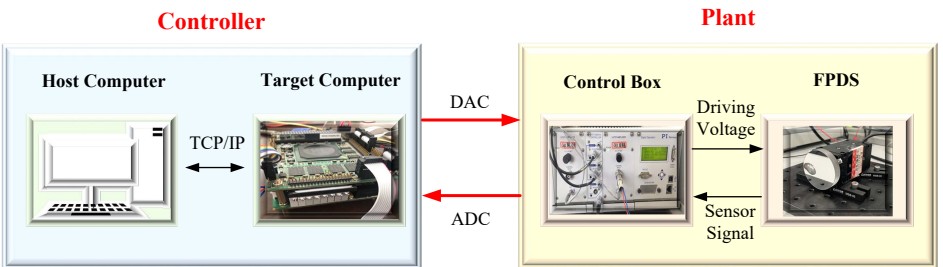

**Figure 13.** Schematic diagram of hardware connection.

*4.2. Parameter Identification*

As illustrated in Figure 4, the proposed WPMBW model cascaded with a linear dynamic model is formulated based on the Hammerstein structure. The identification of Hammerstein-based systems has garnered significant attention [42,50–52]. It has been proved that under the excitation of the PRBS signal, the output of the rate-independent nonlinear component would eventually be stable as the PRBS signal. Therefore, previous works commonly apply the PRBS signal to decouple the Hammerstein model. Additionally, the PRBS signal provides several advantages for system identification, including broad frequency range excitation, good correlation properties, easy implementation, and a non-repetitive pattern. This work employs the PRBS signal to decouple the WPMBW model from the linear dynamic model. The decoupling identification approach for the proposed model consists of the following two steps, as summarized in Figure 14.

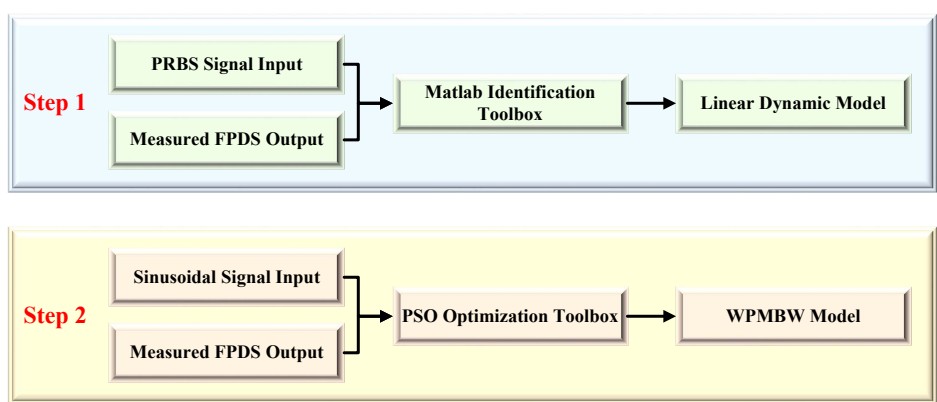

**Figure 14.** Flow chart of the two-step decoupling identification approach for the proposed WPMBW model cascaded with a linear dynamic model.

First step: Identification of the linear dynamic model (8). In order to capture the high-frequency content of the FPDS, the generation period of the PRBS signal is set to 1 ms. The PC104 controller generates an 11-bit PRBS signal with a 4.5 to 5.5 V voltage range. The designed PRBS signal repeats five times to average the measurement noise. Figure 15 depicts the PRBS input and the corresponding FPDS output angles that were measured between 2 and 2.5 s. Considering the presence of a DC offset in the PRBS input, it is necessary to subtract the average value of both the PRBS input and the measured FPDS output angle from their respective signals. Removing the means serves to mitigate the impact of bias. It allows the identification algorithm to focus on the dynamic behavior and variations of the system, which is often the region of interest for modeling system dynamics. Using the identification toolbox of MATLAB, the parameters of the linear dynamic model (8) are identified

as follows: $a_0 = 5.4210 \times 10^{12}$, $a_1 = 1.1830 \times 10^{10}$, $a_2 = 1.0460 \times 10^7$, $a_3 = 8.7530 \times 10^3$, $a_4 = 1$, $b_0 = 5.4210 \times 10^{12}$, $b_1 = -4.0020 \times 10^8$, and $b_2 = 4.0830 \times 10^6$.

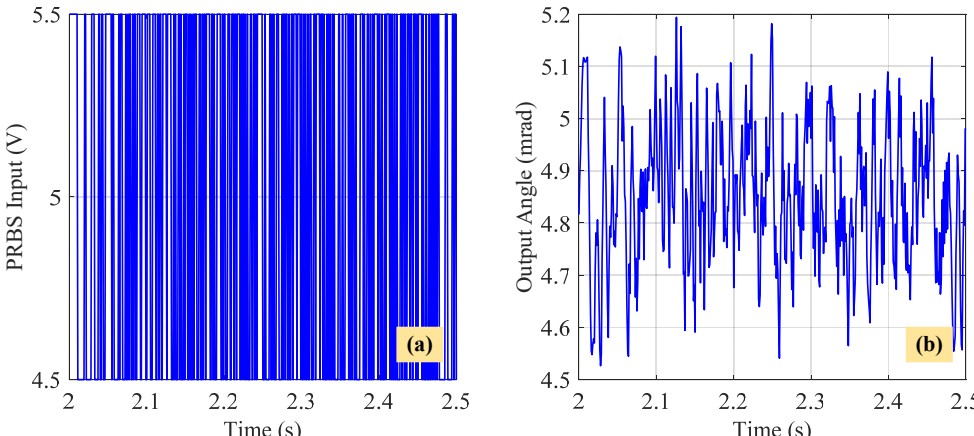

**Figure 15.** The PRBS input and the corresponding FPDS output angles that were measured from 2 to 2.5 s. (**a**) The designed PRBS input from 4.5 to 5.5 V. (**b**) The measured FPDS output angles.

The rate-independent nonlinear component is denoted as $H(\cdot)$ and the rate-dependent linear dynamic component as $G(\cdot)$. It has been demonstrated that any pair of $(\eta H(\cdot), G(\cdot)/\eta)$ where $\eta \neq 0$ will produce an identical output if the input is the same [42]. Therefore, in this work, the gain of the linear dynamic model is fixed to unit to ensure the uniqueness of the identified results.

It is worth noting that the accuracy of the linear model estimation results could be improved by increasing the degrees of the linear dynamic model (*n* and *m*). However, the model complexity also increases, which takes up more computational resources. As a result, there is a trade-off between the model accuracy and complexity. As depicted in Figure 16, the experimental response exhibits a sharp decline followed by an ascent at approximately 180 Hz, resembling the shape of the letter v, indicating that the system model contains a notch filter. Consequently, it can be hypothesized that the system model encompasses a minimum of three poles and two zeros. Through this experiment, it was found that the optimal estimation result was attained when $n = 4$ and $m = 2$. When increasing the values of both *m* and *n* beyond the determined optimal values, the computational complexity escalates accordingly. However, the resulting enhancement in model accuracy is marginal.

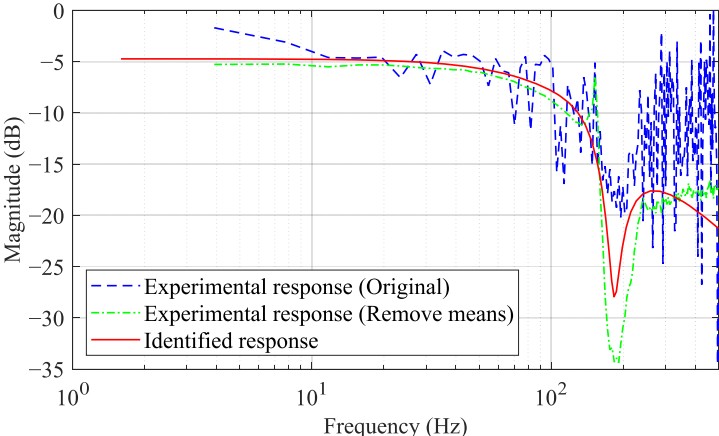

**Figure 16.** Amplitude response of the measured data and the identified linear dynamic model.

Figure 16 also compares the amplitude response of the identified linear dynamic model with the experimental data. It is evident that the identified response exhibits a good fit with the experimental response (remove means) up to 100 Hz. In the high-frequency

range, due to the limited number of orders utilized in our identification model, there exists a certain level of discrepancy between the identified response and the experimental response. However, overall, the identification model still captures the general trend of the experimental response and follows its variations. Moreover, since the frequency of the desired tracking trajectory is under 20 Hz, the identified model can reasonably simulate the dynamic behavior of an FPDS.

Second step: Identification of the WPMBW model (6) and (7). Since the hysteresis loops of an FPDS are amplitude-dependent, a sinusoidal signal with varying amplitudes is applied as follows:

$$u(t) = 5.45e^{-0.21t}[\sin(2\pi t + 1.5\pi) + 1] \text{ V} \tag{19}$$

Given that the input frequency of (19) is only 1 Hz, the effect of the linear dynamic model can be ignored. Based on the training input (19) and measured output angles, the PSO approach is employed to find the best parameters of the WPMBW model. The cost function is defined as follows:

$$F(\rho, \lambda, \mu, c_1, c_2) = \frac{1}{N} \sum_{i=1}^{N} [y(i) - v(i)]^2 \tag{20}$$

where $N$ is the total number of sampling points and $y(i)$ and $v(i)$ denote the $i$th measured output angle of the FPDS and the $i$th simulated output angle of the WPMBW model, respectively.

Figure 17 illustrates the PSO iteration process of the WPMBW parameters. It can be observed that the PSO algorithm terminated at about 1800 epochs because the global minimum value of the cost function was not altered by a magnitude of at least $1 \times 10^{-25}$ for a continuous span of 250 epochs. The identified results of the WPMBW model are $\rho = 0.6485$, $\lambda = 0.7217$, $\mu = 0.7440$, $c_1 = 0.9268$, and $c_2 = 0.0048$.

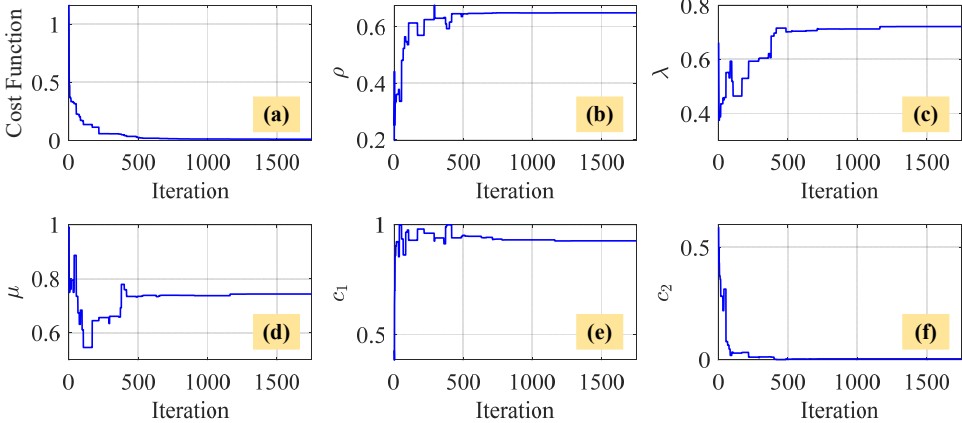

**Figure 17.** The PSO iteration process of the WPMBW parameters. (**a**) The iteration process of the cost function (20). (**b**) The iteration process of $\rho$. (**c**) The iteration process of $\lambda$. (**d**) The iteration process of $\mu$. (**e**) The iteration process of $c_1$. (**f**) The iteration process of $c_2$.

To this end, the parameters of the proposed WPMBW model cascaded with a linear dynamic model have been determined by the two-step decoupling identification approach. In what follows, more experiments are carried out to verify the effectiveness and feasibility of the identified model.

### 4.3. Model Verification

In the following comparative experiments, the CBW model (1) and (2) and the traditional asymmetric BW model in [36] are implemented:

(1)    The CBW model. Based on the training input (19) and measured output angles of the FPDS, the parameters of the CBW model are identified through the PSO algorithm, as

illustrated in Figure 18. The identified parameters of the CBW model are as follows: $\alpha = 0.3554$, $\beta = 0.3485$, $\gamma = 0.2111$, and $d = 0.9248$.

(2) The traditional asymmetric BW model is introduced here as follows:

$$\begin{cases} m\ddot{y}(t) + b\dot{y}(t) + ky(t) = k(d_0 u(t) - h(t)) \\ \dot{h}(t) = \alpha_0 \dot{u}(t) - \beta_0 |\dot{u}(t)||h(t)|^{n_h-1}h(t) - \gamma_0 \dot{u}(t)|h(t)|^{n_h} - \delta u(t)\text{sgn}(\dot{u}(t)) \end{cases} \quad (21)$$

where $m$, $b$, and $k$ represent the mass, damping, and stiffness coefficients of the FPDS, respectively; $d_0$ denotes the piezoelectric coefficient; and parameters $\alpha_0$, $\beta_0$, $\gamma_0$, and $\delta$ are used to regulate the shape of hysteresis loops. In particular, the term $\delta u(t)\text{sgn}(\dot{u}(t))$ accounts for the asymmetric feature of hysteresis loops. It should be mentioned that the original term $k_0/\tau e^{-t/\tau}u(t)$ in [36] has been replaced by $d_0 u(t)$ since the piezoelectric coefficient is a constant [53]. Similar to the identification procedure of the CBW model, the parameters of the traditional asymmetric BW model are acquired as follows: $m = 0.1029$, $b = 81.3370$, $k = 3.1628 \times 10^4$, $d_0 = 0.9221$, $\alpha_0 = 0.3050$, $\beta_0 = 0.5913$, $\gamma_0 = 0.1649$, $\delta = 0.0415$, and $n_h = 1.7170$, as shown in Figure 19.

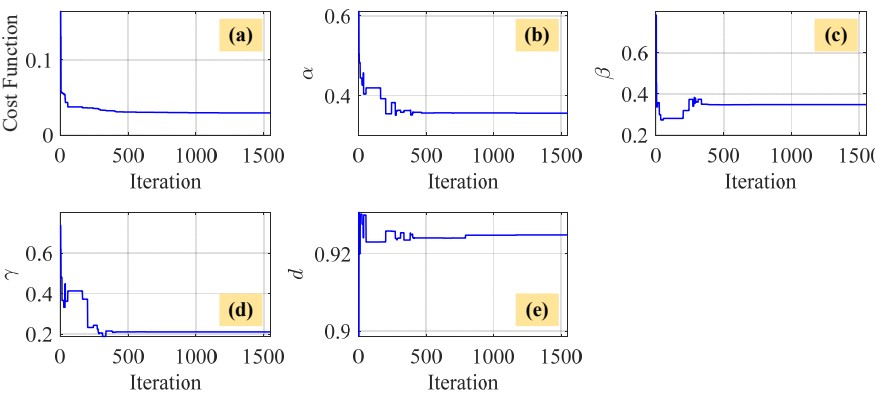

**Figure 18.** The PSO iteration process of the CBW parameters. (**a**) The iteration process of the cost function. (**b**) The iteration process of $\alpha$. (**c**) The iteration process of $\beta$. (**d**) The iteration process of $\gamma$. (**e**) The iteration process of $d$.

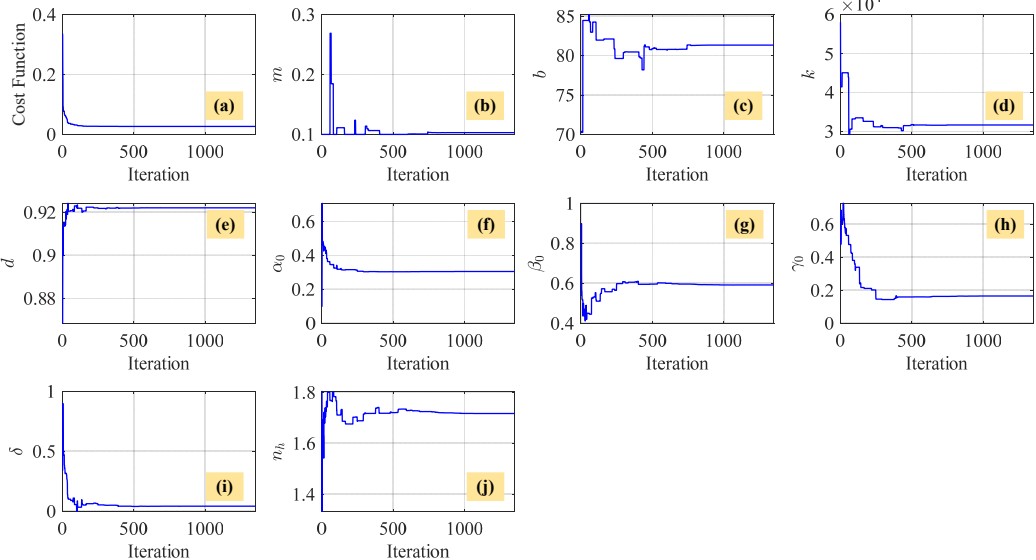

**Figure 19.** The PSO iteration process of the traditional asymmetric BW parameters. (**a**) The iteration process of the cost function. (**b**) The iteration process of $m$. (**c**) The iteration process of $b$. (**d**) The iteration process of $k$. (**e**) The iteration process of $d$. (**f**) The iteration process of $\alpha_0$. (**g**) The iteration process of $\beta_0$. (**h**) The iteration process of $\gamma_0$. (**i**) The iteration process of $\delta$. (**j**) The iteration process of $n_h$.

Figure 20 depicts the identification results of the CBW model, the traditional asymmetric BW model [36], and the proposed WPMBW model cascaded with a linear dynamic model in comparison with the measured training output angles. It can be observed from the magnified plot of Figure 20c that both the model in [36] and the proposed model can describe the asymmetric hysteresis loops. However, the estimated angles of the proposed model match better with the measured angles. To more clearly demonstrate the identification accuracy of each model, the following two performance indices are employed: the maximum percentage modeling error (MPME) and the root-mean-square error (RMSE). They are defined as follows:

$$
\text{MPME(\%)} = \frac{\max_i |y(i) - \hat{y}(i)|}{\max(y) - \min(y)} \times 100\% \tag{22}
$$

$$
\text{RMSE(mrad)} = \sqrt{\frac{1}{N}\left[\sum_{i=1}^{N}(y(i) - \hat{y}(i))^2\right]} \tag{23}
$$

where $y(i)$ and $\hat{y}(i)$ are the $i$th measured and $i$th estimated angles, respectively. The corresponding identification errors are summarized in Table 1. Concerning the MPME, the proposed model achieves the lowest value of 4.58%, followed by the model in [36] with a value of 8.90%, and the CBW model with the highest value of 11.99%. Similarly, the RMSE of the proposed model is 52.17% lower than that of the CBW model and 38.43% lower than that of the model in [36]. Overall, the identification results highlight the superior performance of the proposed model compared to the other two models.

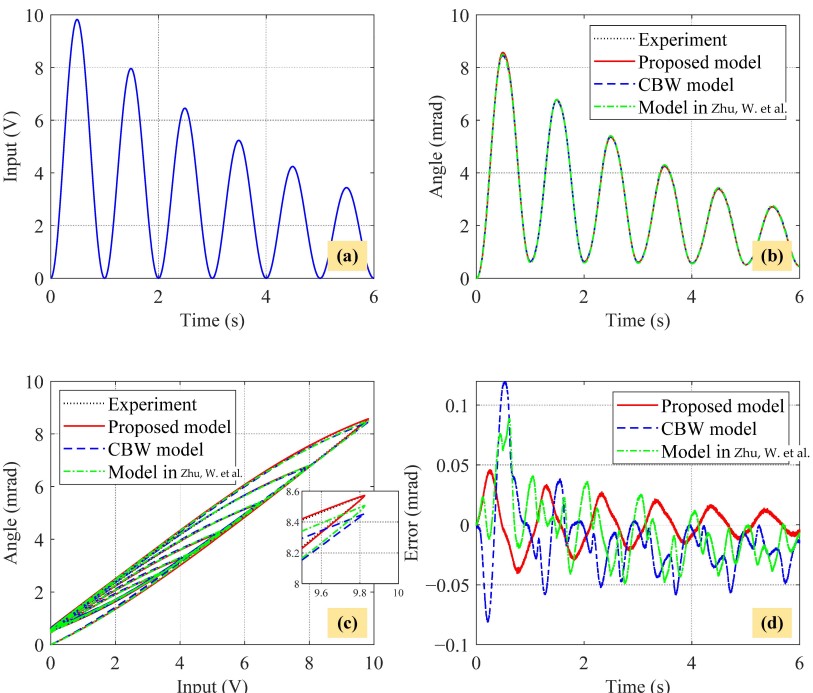

**Figure 20.** Identification results of the CBW model, the traditional asymmetric BW model in [36], and the proposed WPMBW model cascaded with a linear dynamic model. (**a**) Training input (19). (**b**) Ouput angles. (**c**) Hysteresis loops. (**d**) Model output errors.

**Table 1.** Identification errors of the CBW model, the model in [36], and the proposed model.

| Errors | CBW Model | Model in [36] | Proposed Model |
| --- | --- | --- | --- |
| MPME (22) | 11.99% | 8.90% | 4.58% |
| RMSE (23) | 0.0345 | 0.0268 | 0.0165 |

To further evaluate the rate-dependent performance of each identified model, a test input with increasing frequencies is applied as follows:

$$u(t) = 4(1 - cos(2\pi ft)) \ \text{V} \tag{24}$$

where $f = 5, 10, 15,$ and 20 Hz. Figure 21 compares hysteresis loops of the experimental response and three identified models under different input frequencies. The corresponding modeling errors are depicted in Figure 22 and calculated in Table 2. It can be observed that both the MPME and RMSE of the model in [36] and the proposed model are significantly lower than those of the CBW model at all frequencies. For example, at 5 Hz, the MPME of the CBW model is 29.37%, while that of the model in [36] and the proposed model are 6.36% and 2.17%, respectively. The RMSE of the CBW model at the same frequency is 0.1578, while that of the model in [36] and the proposed model are 0.0273 and 0.0113, respectively. Similar observations can be made for other frequencies. These results demonstrate that the CBW model cannot capture the rate-dependent and asymmetric hysteresis loops. Between the model in [36] and the proposed model, it can be seen that the former has bigger MPME and RMSE than the latter at all frequencies. The MPME of the model in [36] ranges from 6.36% to 25.83%, while that of the proposed model ranges only from 2.17% to 15.01%. Regarding the RMSE, the proposed model is 64.00%, 50.38%, 42.98%, and 37.63% less than that of the model in [36] at frequencies of 5 Hz, 10 Hz, 15 Hz, and 20 Hz, respectively. Therefore, the proposed WPMBW model cascaded with a linear dynamic model can well describe the asymmetric and rate-dependent hysteresis loops of an FPDS.

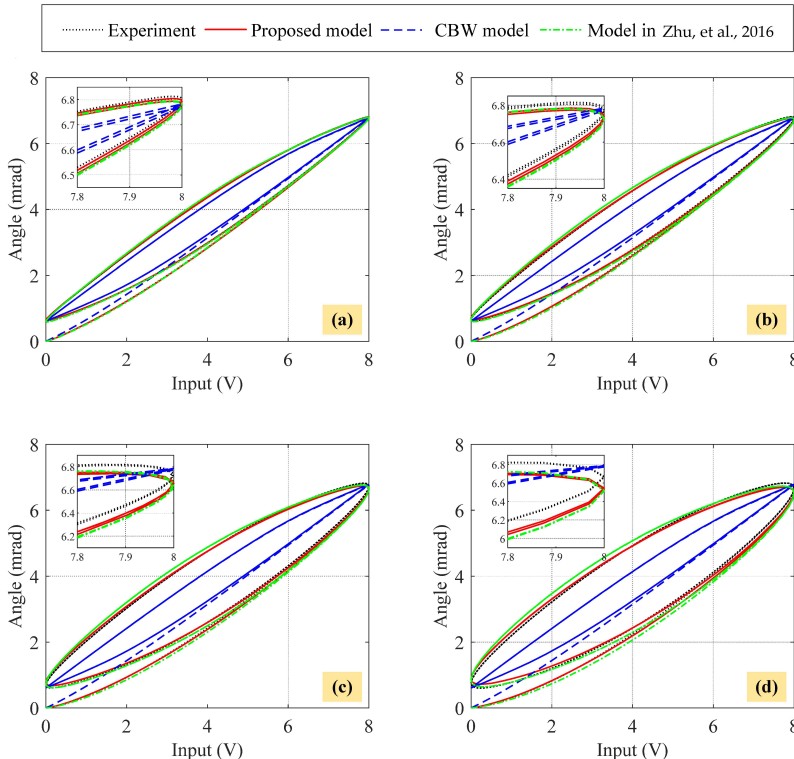

**Figure 21.** Hysteresis loops of the experimental response, the CBW model, the model in [36], and the proposed WPMBW model cascaded with a linear dynamic model under test input (24). (**a**) 5 Hz. (**b**) 10 Hz. (**c**) 15 Hz. (**d**) 20 Hz.

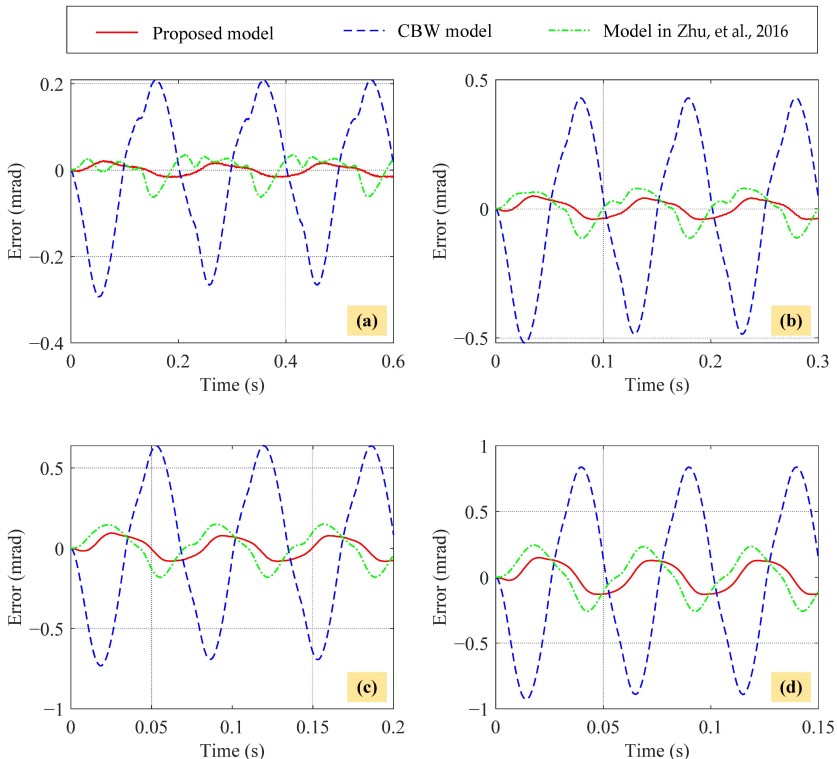

**Figure 22.** Test errors of the CBW model, the model in [36], and the proposed WPMBW model cascaded with a linear dynamic model under test input (24). (**a**) 5 Hz. (**b**) 10 Hz. (**c**) 15 Hz. (**d**) 20 Hz.

**Table 2.** Test errors of the CBW model, the model in [36], and the proposed model.

| Frequency | Errors | CBW Model | Model in [36] | Proposed Model |
|---|---|---|---|---|
| 5 Hz | MPME | 29.37% | 6.36% | 2.17% |
| | RMSE | 0.1578 | 0.0273 | 0.0113 |
| 10 Hz | MPME | 51.91% | 11.39% | 5.03% |
| | RMSE | 0.3073 | 0.0595 | 0.0295 |
| 15 Hz | MPME | 73.14% | 18.07% | 9.56% |
| | RMSE | 0.4506 | 0.1037 | 0.0592 |
| 20 Hz | MPME | 92.96% | 25.85% | 15.01% |
| | RMSE | 0.5886 | 0.1576 | 0.0983 |

*4.4. Hysteresis Compensation*

Furthermore, feedforward hysteresis compensation experiments on an FPDS are carried out to validate the performance of the identified models. The feedforward controllers based on the CBW model and the model in [36] are designed using the IMS and are presented as follows:

$$\begin{cases} u(t) = \frac{1}{d}(y_d(t) + \hat{h}(t)) \\ \dot{\hat{h}}(t) = \alpha \dot{u}(t) - \beta |\dot{u}(t)| \hat{h}(t) - \gamma \dot{u}(t) \left| \hat{h}(t) \right| \end{cases} \quad (25)$$

and

$$\begin{cases} u(t) = \frac{1}{d_0}\left(\frac{1}{k}(m\ddot{y}_d(t) + b\dot{y}_d(t) + ky_d(t)) + \hat{h}(t)\right) \\ \dot{\hat{h}}(t) = \alpha_0 \dot{u}(t) - \beta_0 |\dot{u}(t)| |\hat{h}(t)|^{n_h - 1}\hat{h}(t) - \gamma_0 \dot{u}(t) |\hat{h}(t)|^{n_h} - \delta u(t)\mathrm{sgn}(\dot{u}(t)) \end{cases} \quad (26)$$

where $y_d$ is the desired reference. It is noteworthy that (25) and (26) encounter algebraic-loop issues, as the calculation of $\hat{h}(t)$ requires the value of $u(t)$. Therefore, a constant time

delay is employed to solve this problem. As described in Section 3.4, the proposed WPMBW model has a direct inverse model, circumventing the algebraic-loop issue. The feedforward controller based on the proposed model is given as follows:

$$
\begin{cases}
\dot{u}(t) = \begin{cases} \hat{w}(t)\frac{1-\rho\lambda u(t)+\rho\mu\hat{w}(t)}{\rho}, & \text{if } \hat{w}(t) \geq 0 \\ \hat{w}(t)\frac{1+\rho\lambda u(t)-\rho\mu\hat{w}(t)}{\rho}, & \text{if } \hat{w}(t) \leq 0 \end{cases} \\
\hat{w}(t) = \frac{-c_1+\sqrt{c_1^2+4c_2\hat{v}(t)}}{2c_2} \\
\hat{v}(t) = \frac{1}{b_0}\left(\sum_{i=0}^{n} a_{n-i}y_d^{(n-i)}(t)\right)
\end{cases}
\tag{27}
$$

It should be noted the zero points of (8) are ignored when calculating the inverse linear dynamic model.

It can be observed from Figure 2 that an FPDS cannot return to its zero position after the first period. Therefore, a bias should be added to the given reference $y_r$ to make $y_r$ trackable. However, the bias may lead to a poor transient response at the initial stages of the experiments since an FPDS is a less-damped system. To achieve a smoother initial transient response, the full-pass filter is implemented as follows:

$$
\ddot{y}_d(t) + 2\xi\omega_n\dot{y}_d(t) + \omega_n^2 y_d(t) = \ddot{y}_r(t) + 2\xi\omega_n\dot{y}_r(t) + \omega_n^2 y_r(t)
\tag{28}
$$

where $\xi$ and $\omega_n$ denote the damping ratio and natural frequency, respectively. In this work, $\xi = 1$ and $\omega_n = 100\pi$. When $\zeta = 1$, the full-pass filter is critically damped, resulting in a rapid convergence of the output signal $y_d$ towards $y_r$ without any oscillation. Greater values of $\omega_n$ correspond to faster response times, whereas lower values lead to slower response times. However, if a smoother response with reduced sensitivity to rapid input changes is desired, opting for a lower $\omega_n$ might be more appropriate. Hence, the selection of $\omega_n$ should strike a balance between response time and smooth response. Considering the maximum input frequency of 20 Hz and the desired smooth response during the initial stage, $\omega_n$ is therefore set to $100\pi$. Using (28), $y_d$ always starts from the zero position, avoiding the bias issue and resulting in a better initial transient response. Moreover, $y_d$ eventually coincides with $y_r$ after a short period.

The schematic diagram of the feedforward hysteresis compensation is illustrated in Figure 23. Two typical tracking references are designed as follows:

$$
y_{r1}(t) = 3.45e^{-0.21t}[\sin(2\pi t + 1.5\pi) + 1] + 1 \ \text{mrad}
\tag{29}
$$

$$
y_{r2}(t) = 3.25e^{-0.15t}[\sin(20\pi e^{-1.1t}t + 1.5\pi) + 1] + 1 \ \text{mrad}
\tag{30}
$$

where $y_{r1}$ is the reference with fixed frequency but varying amplitudes, while $y_{r2}$ is the reference with varying amplitudes and frequencies.

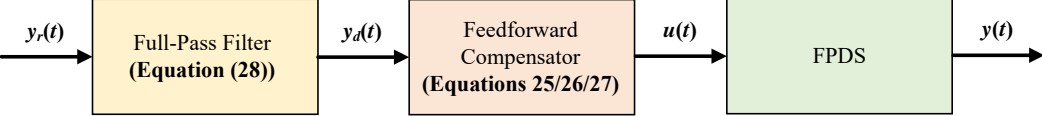

**Figure 23.** Schematic diagram of feedforward hysteresis compensation.

Figure 24 depicts the feedforward hysteresis compensation results based on the CBW model, the model in [36], and the proposed model. The corresponding tracking errors are tabulated in Table 3. The 45° line is introduced to demonstrate the hysteresis compensation performance. Ideally, if the hysteresis is fully compensated, the mapping between desired and measured angles will perfectly align with the 45° line. On the other hand, in the presence of poor hysteresis compensation, the hysteresis loop will widen, leading to deviations from the 45° line. From Figure 24, it is evident that the hysteresis loops of the CBW model exhibit greater width compared to the model in [36] and the proposed model, indicating the poorer hysteresis compensation performance. Regarding tracking errors of

the first desired trajectory (29), the CBW model achieves an MPME of 24.10%, which is 9.23% and 10.25% bigger than that of the model in [36] and the proposed model, respectively. Similarly, in terms of the RMSE, the CBW model exhibits a value of 0.0366, while the model in [36] achieves a lower RMSE of 0.0227, and the proposed model demonstrates the best performance with an RMSE of 0.0111. Since the CBW model cannot describe the asymmetric and rate-dependent hysteresis loops, the hysteresis compensation results continue to deteriorate for the second desired trajectory (30), where the MPME and RMSE are 44.70% and 0.0608, respectively. In contrast, the proposed model offers the lowest MPME of 13.78%, which is 30.92% and 4.03% smaller than that of the CBW model and the model in [36], respectively. Meanwhile, the RMSE of the proposed model is 79.44% less than that of the CBW model and 38.73% less than that of the model in [36]. Therefore, experimental results demonstrate the effectiveness and superiority of the proposed model in hysteresis modeling and compensation.

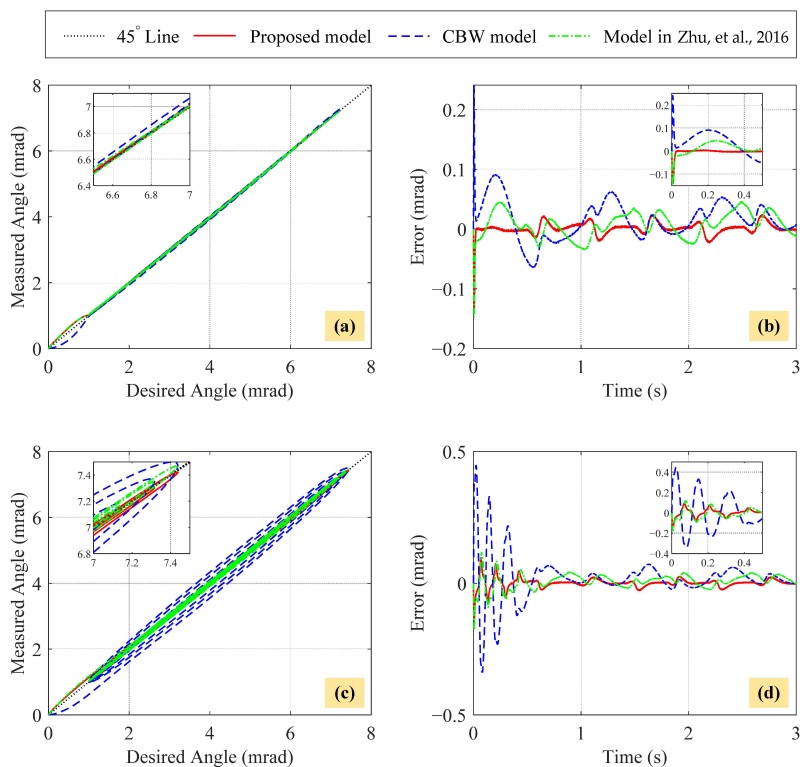

**Figure 24.** Tracking results of different feedforward hysteresis compensators based on the CBW model, the model in [36], and the proposed model. (**a**) Hysteresis reduction under the desired reference (29). (**b**) Tracking errors under (29). (**c**) Hysteresis reduction under the desired reference (30). (**d**) Tracking errors under (30).

**Table 3.** Feedforward hysteresis compensation errors of the CBW model, the model in [36], and the proposed model.

| References | Errors | CBW Model | Model in [36] | Proposed Model |
|:---:|:---:|:---:|:---:|:---:|
| (29) | MPME | 24.10% | 14.87% | 13.85% |
| | RMSE | 0.0366 | 0.0227 | 0.0111 |
| (30) | MPME | 44.70% | 17.81% | 13.78% |
| | RMSE | 0.0608 | 0.0204 | 0.0125 |

### 4.5. Discussion

According to the experiments mentioned above, it is convincing that the proposed WPMBW model cascaded with a linear dynamic model offers higher modeling accuracy

than the existing CBW and traditional asymmetric BW models [36]. The primary factor behind this improvement is the incorporation of the weighted polynomial function and the linear dynamic model, which effectively capture the asymmetric and rate-dependent hysteresis characteristics, respectively. The MBW model can directly describe the counterclockwise hysteresis loops with fewer parameters than the CBW model. Furthermore, the MBW model provides a direct inverse model, which solves the algebraic-loop problems encountered in the inverse CBW model. The developed two-step decoupling identification approach simplifies the identification process. Specifically, only five parameters are required to be adjusted by the PSO toolbox. The feedforward hysteresis compensators based on the three identified models are implemented using (25)–(27). It is worth mentioning that the inverse linear dynamic model is disregarded when the desired signal is not smooth. In this work, the desired signal is predefined to evaluate the hysteresis compensation performance of the identified models.

Since the hysteresis behavior is effectively compensated, the FPDS can be treated as a linear system with perturbations. For linear systems, there are various feedback control techniques to enhance tracking performance, such as PID control, adaptive control, and sliding mode control, just to name a few.

Due to the ongoing development of the UAV image stabilization platform based on an FPDS, this research cannot provide the image effects after disturbance compensation. However, existing studies, such as in [44], demonstrate the successful implementation of voice coil actuators for drone camera stabilization. Considering the higher positioning resolution and open-loop bandwidth offered by an FPDS, it can be anticipated that they will likewise achieve favorable outcomes.

## 5. Conclusions

To meet the challenge of the UAV image stabilization system, a novel WPMBW model cascaded with a linear dynamic model is presented to describe the counterclockwise, asymmetric, and rate-dependent hysteresis loops of an FPDS in this paper. The weighted polynomial function is utilized to capture the asymmetric characteristic, while the linear dynamic model is used to specify the rate-dependent feature. Compared to the CBW model, the MBW model requires fewer parameters and provides a direct inverse model, circumventing the algebraic-loop problem. The relationship between the parameters of the WPMBW model and the shape of hysteresis loops is also investigated, including the derivation of the output bound of the MBW model. Inspired by the Hammerstein structure, a two-step decoupling identification methodology is formulated. First, the PRBS signal is employed to identify the linear dynamic model, and the parameters of the linear dynamic model are predicted through the MATLAB identification toolbox. Second, the sinusoidal signal is applied to stimulate the hysteresis phenomenon, and the parameters of the WPMBW model are estimated by the PSO toolbox. Comparative experimental results demonstrate that the proposed model outperforms the CBW and traditional asymmetric BW models in modeling accuracy and hysteresis compensation. In the future, FPDS will be integrated into the UAV image stabilization system to compensate for disturbances during the flight.

**Author Contributions:** Conceptualization, J.L.; methodology, J.L.; software, J.L.; validation, J.L. and J.W.; formal analysis, J.L. and J.W.; investigation, J.L., J.W., Y.B. and X.Z.; resources, J.W.; data curation, J.L.; writing—original draft preparation, J.L.; writing—review and editing, J.W., Y.B. and X.Z.; supervision, J.W. and Y.B.; funding acquisition, J.W. All authors have read and agreed to the published version of the manuscript.

**Funding:** This work was supported in part by the Defence Science and Technology Foundation of China under Grant No. 173 (2021-JCJQ-JJ-0883).

**Institutional Review Board Statement:** Not applicable.

**Informed Consent Statement:** Not applicable.

**Data Availability Statement:** Not applicable.

**Conflicts of Interest:** The authors declare no conflict of interest.

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
