# Peer review of "Hysteresis Modeling and Compensation for a Fast Piezo-Driven Scanner in the UAV Image Stabilization System"

_drones, doi:10.3390/drones7060392_

Round 1

Reviewer 1 Report

The paper is well written. Conclusions, descriptions, analyses and results are sound. I have a few suggestions:

1) In the abstract: "As the proposed model follows the Hammerstein structure, the pseudorandom binary sequence (PRBS) input is employed to decouple..." -> it is not obvious that the PRBS should be used due to the usage of a Hammerstein structure. In the abstract, there is no space for such an explanation. I suggest to eliminate the mention to the Hammerstein structure.

2) p. 2, line 74: the first time that "algebraic-loop problem" is cited, the problem should be clearly stated. This explanations is made only in a later section

3) p. 13, line 309: why these values of m and n? Please justify the choice.

4) p. 19, line 403: "zeta = 1 and wn = 100\pi" -> Again: why these values? Please justify the choice.

Author Response

We sincerely thank you for the invaluable feedback and comments that helped to improve the quality of our submission. Following your comments, we have conducted a comprehensive revision of our manuscript. More details can be found in point-to-point responses. We hope that with the changes made in the revised version, it can now be acceptable in the journal.

Reviewer 2 Report

Dear Authors,

The manuscript should be revised before it is published. My comments are in the attachment.

Kind Regards

Quality of the language is fine. 

Author Response

(The authors gave the same response as above.)

Round 2

Reviewer 2 Report

Dear Authors,

I have no further question related to the revised version of the paper. It can be published after minor revision (minor methodological errors and text editing).

Kind Regards

Minor editing of english is required.